# Effect of Slag on the Strength and Shrinkage Properties of Metakaolin-Based Geopolymers

**DOI:** 10.3390/ma15082944

**Published:** 2022-04-18

**Authors:** Jianghuai Zhan, Hongbo Li, Qun Pan, Zhenyun Cheng, Huang Li, Bo Fu

**Affiliations:** 1College of Civil and Hydraulic Engineering, Ningxia University, Yinchuan 750021, China; zhanjianghuai2020@163.com (J.Z.); 13469578416@nxu.edu.cn (H.L.); 2Chongqing Construction Science and Research Institute Co., Ltd., Chongqing 400060, China; panqunzb@163.com; 3School of Civil Engineering, North Minzu University, Yinchuan 750021, China; 2016035@nmu.edu.cn; 4China Construction Fifth Engineering Bureau Co., Ltd., Changsha 410011, China; lihuang1989@hnu.edu.cn; 5National Energy Group, Coal Chemical Industry Technology Research Institute, Ningxia Coal Industry Co., Ltd., Yinchuan 720021, China

**Keywords:** alkali-activated metakaolin-based geopolymer, slag content, Na_2_O content, compressive strength, volume stability

## Abstract

Metakaolin-based geopolymers possess excellent corrosion and high-temperature resistance, which are advantageous compared to ordinary Portland cement. The addition of slag in metakaolin-based geopolymers is a promising approach to improve their mechanical properties. Thus, this study investigated the effect of slag content on the strength and shrinkage properties of metakaolin-based geopolymers. Increasing the slag content and Na_2_O content was beneficial to the reaction of alkali-activated metakaolin-based geopolymers, thereby improving their compressive strength and density. After 56 days of aging, a maximum compressive strength of 86.1 MPa was achieved for a metakaolin-based geopolymer with a slag content of 50 mass%. When the Na_2_O content was 12%, the compressive strength of the metakaolin geopolymers with a slag content of 30% was 42.36% higher than those with a Na_2_O content of 8%. However, as the slag and alkali contents increased, the reaction rate of the metakaolin-based geopolymers increased, which significantly decreased the porosity, increased the shrinkage, and decreased the volumetric stability of the system. In this paper, in-depth study of the volume stability of alkali-activated metakaolin-based geopolymers plays an important role in further understanding, controlling, and utilizing the deformation behavior of geopolymers.

## 1. Introduction

Geopolymers are Si–O–Al network-based dense cementitious materials that are formed by the ionic and covalent bonding between active silicate materials (such as metakaolin and fly ash) under the catalysis of alkali activators (such as caustic soda and alkali-containing silicates) [1,2]. Alkali-activated metakaolin-based geopolymers are a new type of cementitious material with a zeolite-like structure formed by metakaolin under an alkali activator with a typical Si/Al ratio of 1:3. Compared with ordinary Portland cement, alkali-activated metakaolin-based geopolymers are more environmentally friendly with excellent corrosion and high-temperature resistance [2,3,4,5]. However, their widespread application in civil engineering is limited due to several disadvantages, such as a long setting time, low early compressive strength, large shrinkage during drying, and potential cracking risks [6]. Accordingly, methods for improving the compressive strength and volume stability of metakaolin-based geopolymers have attracted global research attention.

To date, scholars have achieved satisfactory results by studying the effect of room-temperature curing [7], increasing the curing temperature [8,9], changing the sodium silicate modulus (SiO_2_/Na_2_O molar ratio) [10], and adding aggregates [11], among other cementitious materials (such as fly ash) [12] on the compressive strength and other properties of metakaolin-based geopolymers. Among these, slag is often used to improve the properties of metakaolin-based geopolymers. Alkali-activated slag-based cementitious materials have the advantages of fast setting and hardening, good early strength [13], and excellent corrosion resistance [14]. The combination of slag and metakaolin can increase the Ca and Si contents in the geopolymer system, which is conducive to the formation of sodium silicate aluminate hydrate (N-A-S-H) and calcium aluminosilicate hydrate (C-A-S-H) hydration products. These two hydration products are interwoven, which increases the compactness of the metakaolin-based geopolymer structure, thereby improving the compressive strength of the system [15].

During setting and hardening, metakaolin-based geopolymers exhibit shrinkage deformation [16]. The shrinkage testing and theoretical research are more complex than investigating the mechanical properties of metakaolin-based geopolymers. Numerous studies have found that the mixture ratio (alkali activator content [17,18], slag content [18], fly ash content [19], and water–binder ratio [20]), curing method [17,21], and additives) significantly influences the volume stability of metakaolin-based geopolymers [22]. Fu et al. [23] reported that metakaolin can coarsen the pore structure of hardened slag, thereby reducing autogenous and drying shrinkage. In addition, increasing the activator concentration and silica modulus increased the autogenous and drying shrinkage of alkali-activated metakaolin-containing slag systems. Si et al. [24] found that the mechanical properties of the metakaolin-based geopolymer sample can be improved by introducing a small amount of glass powder (5–10 wt.%). The added glass powder reduced the water loss rate of the samples under the drying conditions, thereby reducing the drying shrinkage at early ages. Wang et al. [25] studied the effect of the silane coupling agent on the properties of the metakaolin base polymer, and found that the addition of the appropriate amount of silane coupling agent could reduce the shrinkage of the sample at high temperatures. In particular, when the sample with 0.2 wt.% silane coupling agent was exposed to 200 °C, the compressive strength reached 65.1 MPa, which was 10.2% higher than that of pure metakaolin base polymer. Dumitru et al. [26] studied the coal-ash base polymer with phosphoric-acid-activated tailings synthesized at room temperature. The acid-activated geopolymers exhibited similar behavior to that of alkali-activated geopolymers during heating. Ribeiro et al. [27] found that increasing the solid–liquid ratio reduced the shrinkage of red mud + metakaolin-based geopolymer concrete. Castel et al. [28] found that after 1 day of aging at 40 °C, the drying shrinkage of low-calcium fly ash geopolymers doped with slag was 1920 με, where με is the unrestrained axial length change per unit length. When the drying temperature was increased to 80 °C, the drying shrinkage decreased to 400 με. However, the effect of curing temperature on the drying shrinkage was obviously reduced after 3 days. In addition to the effect of temperature, increasing the curing humidity can reduce the drying shrinkage. When the relative humidity was increased from 50% to 99%, the drying shrinkage decreased by 3.8 times [29]. Further, after adding 8 wt.% shrinkage additive to a slag-based geopolymer concrete, the drying shrinkage decreased from 420 to 190 με [30]. Fang et al. [31] hypothesized that the autogenous shrinkage of geopolymers is caused by the continuous recombination and rearrangement of the aluminosilicate gel structure.

Although the strength and volume stability of metakaolin-based geopolymers have been extensively studied, to date, only a few studies have studied both these properties. Therefore, in this study, the strength and volume stability of metakaolin-based geopolymers were studied according to the slag and Na_2_O content (mass fraction). To clarify the influence of the slag and Na_2_O content on the compressive strength and shrinkage performance of alkali-activated metakaolin-based geopolymers, the hydration products and pore structure were studied through Fourier transform infrared spectroscopy (FT-IR), scanning electron microscopy (SEM), and mercury intrusion porosimetry (MIP). The findings of this study provide a theoretical basis for the preparation of metakaolin-based geopolymers materials with good volume stability.

## 2. Materials and Methods

### 2.1. Raw Materials

#### 2.1.1. Metakaolin and Slag

In this study, metakaolin, which is the amorphous aluminum silicate formed from ultrafine kaolin calcined at 600–900 °C, and blast furnace slag, hereafter referred to as “slag,” were sourced from Hohhot Mongolia Chaopai Co., Ltd. (Hohhot, China), and Ningxia Iron and Steel Group (Zhongwei, China), respectively. The main chemical components of these materials are listed in Table 1, and their particle size distribution is shown in Figure 1.

#### 2.1.2. Experimental Sand

Standard sand produced by Xiamen Aisiou Standard Sand Co., Ltd., Xiamen, China, was used to prepare the geopolymers following the GB/T 17671-1999 Cement Mortar Strength Test Method specification [32], and the size range of the standard sand was 0.25–0.5 mm. The specific gravity, which is the ratio of the density of the substance at its fully dense state to that at the standard atmospheric pressure and pure H_2_O at 3.98 °C (999.972 kg/m^3^), and the fineness modulus, which is a dimensionless indicator of the thickness and grain diameter, of the sand were 1.41 and 2.3, respectively.

#### 2.1.3. Alkali Activator

A sodium silicate solution (waterglass; modulus (SiO_2_/Na_2_O molar ratio) of 3.38; containing 28.25 wt.% SiO_2_, 8.63 wt.% Na_2_O, and 63.12 wt.% H_2_O) obtained from Shandong Quansheng Chemical Technology Co., Ltd., Weifang, China, was used in the study. Sodium hydroxide in its flake form (purity: 98%) was produced by Chengdu Puhe Chemical Co., Ltd., Deyang, China. Deionized water was used in the study.

### 2.2. Sample Preparation

The activators used in this study were prepared by blending sodium hydroxide, waterglass solution, and deionized water 24 h before the sample manufacturing. A constant water-to-binder (where water refers to the deionized water and binder is metakaolin + slag) mass ratio of 0.45 was adopted for preparing the pastes. To investigate the effect of slag, the total Na_2_O content in sodium silicate and sodium hydroxide and modulus (SiO_2_/Na_2_O molar ratio) in the activator were kept constant at 10 and 1.5 mass% of the binder, respectively. In addition, two activators with 8 and 12 mass% Na_2_O with a modulus of 1.5, which exhibited a good activation efficiency on alkali-activated metakaolin-based geopolymers according to a previous study [23], were prepared to study the effects of the activator. The binder–sand ratio (wt.%) was set to 1:2. Table 2 shows the mixture proportion of the samples.

The mortar strength tests of the geopolymers were carried out according to ISO GB/T 17671-1999 [32]. For the preparation of the alkali-activated metakaolin-based geopolymer paste, metakaolin and slag were placed in a mixing pot and mixed for 1 min. Subsequently, the activator was added and stirred for 2 min until a uniform mixture was obtained. For the preparation of the mortar, an additional time of 2 min was needed when standard sand was added. The mortar was then cast into steel molds with the dimensions of 4 cm × 4 cm × 4 cm and 4 cm × 4 cm × 16 cm, and covered with a plastic film for 24 h. The samples were demolded and placed in a standard curing chamber with a constant temperature of 20 ± 2 °C and relative humidity of >95% until the tests.

### 2.3. Test Methods

#### 2.3.1. Test Design

The design idea and implementation steps of the experiment conducted in this study are shown in Figure 2.

#### 2.3.2. Mechanical Properties

The mortar strength tests of the geopolymers were carried out according to ISO GB/T 17671-1999 [32]. The geopolymer mortar samples were stirred and poured into a steel mold with the dimensions of 4 cm × 4 cm × 4 cm and vibrated to discharge the bubbles. The bubbles were placed in a standard curing box and maintained at a temperature of 20 ± 2 °C and relative humidity of >90% for 24 h. The resulting paste was removed from the mold and maintained until the desired test age. The compressive strength of the cured samples was obtained using a computer-controlled constant-stress pressure testing machine.

#### 2.3.3. Shrinkage Performance

The mortar drying shrinkage tests of the geopolymers were conducted according to JC/T 603-2004 [33]. A 4 cm × 4 cm × 16 cm test mold was used (the detailed process is shown in Section 2.2). After molding, the specimen was placed in a standard curing environment, maintained for 24 h, and removed from the mold. The specimen was then allowed to cure at a relative humidity of 80% and temperature of 25 °C until reaching the desired test age. The test results were taken as the average of three measurements on separate specimens.

The autogenous shrinkage tests were conducted using a CABR-ZES non-contact bellow according to ASTMC 1698-2009 (2014) [34]. The inner diameter and length of the bellows were 29 and 430 mm, respectively. The prepared metakaolin-slag mixed paste was loaded into the bellows, and the bubbles inside the paste were eliminated by vibration. Subsequently, the bellows were sealed with plastic plugs to prevent water loss. Finally, the bellows were placed on a special support to store the sample at a constant temperature of 23 ± 1 °C. Automatic measurements and sampling were carried out at 1-min intervals. Three samples were tested in parallel, and the average values were used to calculate the autogenous shrinkage value.

The chemical shrinkage tests were carried out according to ASTM-C1608 [35], following the sealed container absolute volume method in a water bath with a constant temperature of 23 °C. The stirred paste was poured into a 250-mL wide-mouthed bottle. The height of the paste was approximately 2 cm. Subsequently, the wide-mouthed bottle was filled with water and sealed using a rubber plug with a capillary measurement tube, which has a measurement range of 2 mL. The chemical shrinkage was determined by measuring the liquid level height in the capillary tube.

#### 2.3.4. Microstructural Analysis

The geopolymer paste samples were prepared according to the proportions in Table 2. After reaching the desired test age, the samples were crushed and placed in ethanol for 3 days to terminate hydration. Subsequently, the samples were dried in a vacuum drying chamber at 60 °C for 24 h. Finally, the FT-IR, SEM, and MIP pore structure analyses were conducted. FT-IR was performed using a Tensor 27 infrared spectrometer (Bruker Corp., Billerica, MA, USA; test range: 4000–400 cm^−1^; resolution: 2 cm^−1^). The pore structure was analyzed using an AutoPore IV 9510 automated mercury porosimeter (Micromeritics Inst. Corp., Norcross, GA, USA; measuring range: 5 nm to 0.34 mm). Finally, the micromorphology was examined using a Quanta 200 scanning electron microscope (FEI Co., Hillsboro, OR, USA; resolution: 5 nm; magnification: 20–10,000×).

## 3. Results and Discussion

### 3.1. Compressive Strength

Figure 3a shows the effect of the slag content on the compressive strength of the alkali-activated metakaolin-based geopolymer test blocks. After 56 days of aging, the geopolymer block with 50 mass% slag had the highest compressive strength of 86.1 MPa. Compared to the blocks with 30, 10, and 0 mass% slag, the block with 50 mass% slag exhibited an improved compressive strength by 0.73%, 14.29%, and 22.80%, respectively. As the slag content increased, the compressive strength of the geopolymer blocks increased, consistent with the conclusions of Paulo et al. [36]. This is attributed to the increase in the introduced CaO content with the slag content, and the easier dissociation and dissolution of the Ca–O bonds than the Si–O and Al–O bonds in alkaline environments. Thus, the large CaO content increased the Ca^2+^ concentration in the paste, which enhanced the dissolution rate and degree of the aluminosilicate minerals in the system and accelerated the hydration reaction. After introducing slag into the metakaolin system, the reaction products were coexisting N-A-S-H and C-A-S-H gels, which were interwoven and intergrown, thereby filling the gaps between the different reaction phases and unreacted particles. This improved the pore size distribution, reduced the system porosity, and increased the density and uniformity of the structure. Consequently, increasing the slag content is conducive to the improvement of the compressive strength of the geopolymer test block. However, Cui et al. [10,11] obtained a different conclusion. Cui studied the influence of different slag contents (0, 20, 40, 60, 80 and 100 mass%) on the compressive strength of metakaolin-based geopolymer. It was found that the compressive strength of metakaolin-based geopolymer with 40% slag content was the greatest at 28 days.

The compressive strength of the metakaolin-based geopolymers with 50 mass% slag was not significantly different from that of the geopolymers with 30 mass% slag. Therefore, in the following discussion, the metakaolin-based geopolymers with 30 mass% slag were selected for the analysis. Figure 3b shows the effects of the Na_2_O content on the compressive strength of the alkali-activated metakaolin-based geopolymer test blocks. For the samples with the same curing time, the compressive strength increased as the Na_2_O content increased. After 3 and 56 days of curing, the compressive strength of the S30-12 geopolymers was 82.42% and 42.36% higher, respectively, than that of the S30-8 geopolymers at the same age. As the curing time increased, the strength of the geopolymers increased rapidly and gradually, consistent with the results of previous studies [37,38]. This is ascribed to the increased Na_2_O content, and dissolution rate of Ca, Si, and Al, which are conducive to the reconstruction and reaction of metakaolin-based geopolymers, thereby increasing the compressive strength.

### 3.2. Drying Shrinkage

Figure 4a shows the effect of the slag content on the drying shrinkage of the alkali-activated metakaolin-based geopolymer specimens. For all specimens, the drying shrinkage increased rapidly in the first 28 days of aging, and then gradually stabilized. As the slag content increased, the drying shrinkage of the geopolymers increased. At 28 days of aging, the drying shrinkages of the S0-10, S10-10, S30-10, and S50-10 geopolymers were 282.47, 441.41, 672.69, and 890.08 × 10^−6^, respectively. After 180 days of aging, the shrinkage of the specimen with 0 mass% slag was 24.74%, 50.19%, and 64.78% lower than that of the specimens with 10, 30, and 50 mass% slag, respectively. Thus, the shrinkage of the slag system is significantly higher than that of the metakaolin system, which is consistent with the results of Fu et al. [14,23,39]. These findings are ascribed to the increase in Ca^2+^ as the slag content increased because Ca^2+^ promotes the formation of C-A-S-H gel, which has viscoelastic/viscoplastic behavior and low creep modulus. Furthermore, the combination of the alkali ions reduces the regularity of the accumulation structure, making the geopolymers more prone to collapse and redistribution during drying [14,40]. Consequently, the associated drying shrinkage is larger with a higher slag content.

Figure 4b shows the similar drying shrinkage behaviors of the metakaolin-based geopolymer specimens with different Na_2_O contents. A large amount of shrinkage occurred in the initial 28 days of aging, after which the shrinkage rate decreased until reaching an approximately constant degree of shrinkage. After 180 days of aging, mortars S30-8, S30-10, and S30-12 had the drying shrinkage values of 334.17 × 10^−6^, 783.13 × 10^−6^, and 1081.67 × 10^−6^, respectively. The shrinkage increased significantly as the Na_2_O content increased, which is consistent with the results of Li et al. [41]. The Na_2_O content greatly affects the proportion and morphology of the generated C-A-S-H and N-A-S-H gels [36]. A high Na_2_O content is not conducive to slag hydration, whereas a low Na_2_O content hinders metakaolin hydration. With a suitable Na_2_O content, the generated C-A-S-H gel is interwoven with the N-A-S-H network structure, which increases the density of the matrix and reduces shrinkage. When the Na_2_O content was high (it was 12%) a fast hydration reaction was noted, resulting in the precipitation of the gel at an early stage, which hindered the subsequent hydration reaction, and consequently, increased the porosity and shrinkage [42]. When the Na_2_O content was 8 mass%, the geopolymers exhibited good volume stability.

Notably, the drying shrinkage curves of the metakaolin-based geopolymers are composed of the expansion during initial curing and followed by shrinkage, which is consistent with the results of Yang et al. [19]. This early-stage expansion can be attributed to the water vapor in the curing box, which cannot be fully maintained in the specimen because of the restrictions of the steel mold for the first 24 h. After demolding, the test block came in full contact with the water vapor, resulting in a more prominent hydration reaction, thereby increasing the specimen length.

### 3.3. Autogenous Shrinkage

Figure 5a shows the effect of the slag content on the autogenous shrinkage behavior of the alkali-activated metakaolin-based geopolymer specimens. Different slag contents resulted in different autogenous shrinkage behaviors. The expansion behavior and strain of mortars S10-10 and S0-10 increased gradually. When the slag content was more than 10 mass%, shrinkage was noted. However, these results are inconsistent with those of previous studies [43,44]. Most studies have concluded that the autogenous shrinkage of geopolymers increases as the slag content increases. Conversely, in this study, the autogenous shrinkage was only noted with a large slag content. When the slag content was small, the geopolymers expanded. Therefore, it is important to study the effects of the slag content on the autogenous shrinkage of alkali-activated metakaolin-based geopolymers for further utilization of metakaolin-based geopolymers.

This discrepancy can be ascribed to the low slag content (0 and 10 mass%). In the alkali excitation process, the alkali reacts with the metakaolin and slag particles, which decreases the ion concentration in the pore solution, thereby increasing the internal relative humidity and expansion of the specimen. As the zeolite-like hydration products generated by the metakaolin hydration reaction have a three-dimensional network structure, the specimen expands with their formation [45]. Palumbo and Melo Neto et al. [46,47] noted the early expansion of metakaolin-based geopolymers. Yang [19] found that with an increase in age, the autogenous shrinkage behavior of a metakaolin-based geopolymer paste with a water–binder ratio of 0.5 was divided into the autogenous expansion during initial curing and subsequent shrinkage strain. Moreover, as the metakaolin content increased, the geopolymer paste expansion became more prominent. When the metakaolin content was 100 mass%, the geopolymer paste exhibited autogenous expansion in the first 28 days of curing, after which the paste began to shrink. When the metakaolin content was 90%, the geopolymer paste expanded because of the autogenous shrinkage in the first 14 days of curing and gradually shrank thereafter. Consequently, as the metakaolin content decreased, the initial expansion stage of the metakaolin-based geopolymer paste shortened, and the later shrinkage stage increased. Similarly, Francesca et al. [48] studied the early volume change in metakaolin-based geopolymers. A metakaolin-based geopolymer with a Si/Al ratio of 1.435 expanded in the first 35 days of curing, whereas a geopolymer with a Si/Al ratio of 1.75 expanded in the first 17 days only, and shrank thereafter. A geopolymer with a Si/Al ratio of 1.99 expanded during aging for 1–2 days and shrank after. The shrinkage increased as the curing age increased. These results are related to the zeolite-like three-dimensional network structure generated by the hydration reaction of the metakaolin-based geopolymers. Additionally, the low geopolymers formed in the early stage greatly influenced the volume expansion [45], consistent with the results of Wild et al. [49]. Melo Neto et al. [47] hypothesized that early polymerization would drive water in the pores to areas with significant water shortage, which would increase the local humidity, decrease the capillary pressure, and start specimen expansion. Herein, when the slag content was increased from 30 to 50 mass%, the extent of autogenous shrinkage increased, consistent with the results of previous studies [7,50]. This occurs because the microstructure of the hydration phase is more sensitive to slag content changes at high slag contents, and the structural densification increases the effective stress caused by the capillary pressure in the saturated pores, thereby promoting autogenous shrinkage.

Figure 5b shows the effect of the Na_2_O content on the autogenous shrinkage of the alkali-activated metakaolin-based geopolymer specimens. As the Na_2_O content increased, the extent of autogenous shrinkage of the geopolymers (largest to smallest) is as follows: S30-12 > S30-10 > S30-8. At 14 days of aging, the autogenous shrinkage of the geopolymers with 12 mass% Na_2_O was 121.32% and 29.41% higher than that of that of specimens with 10 and 8 mass% Na_2_O, respectively. As the Na_2_O content increased, the extent of autogenous shrinkage increased because of the increased hydration rate of the geopolymers. As the hydration progressed, the degree of paste hydration increased, the capillaries formed became thinner, the relative humidity decreased, and the surface tension of the pore solution increased, resulting in the increased autogenous shrinkage [37,51]. However, as the curing age increased, the extent of late shrinkage decreased gradually. The largest shrinkage rate for the geopolymer was obtained at 2–3 days. Thereafter, the shrinkage rate decreased gradually. With the Na_2_O contents of 8, 10, and 12 mass%, the autogenous shrinkage ratio after 14 days decreased by 27.73%, 24.3%, and 23.98%, respectively, compared with that after 3 days. According to previous studies, metakaolin-based geopolymer pastes undergo pore water discharge after molding [52], which mitigates autogenous shrinkage. Metakaolin can promote the crystallinity of the clinocalcite phase, which also plays a role in limiting shrinkage [53].

### 3.4. Chemical Shrinkage

Figure 6a shows the effect of the slag content on the chemical shrinkage of the alkali-activated metakaolin-based geopolymer specimens. During curing, the geopolymer paste expanded at 0–12 h and shrank after 12 h. Thereafter, the extent of chemical shrinkage greatly increased until 14 days and subsequently stabilized. The chemical shrinkage of the geopolymers (largest to smallest) is as follows: S50-10 > S30-10 > S10-10 > S0-10. At 24 h, the chemical shrinkage of the S50-10, S30-10, S10-10, and S0-10 geopolymers were 0.017, 0.011, 0.010, and 0.0033 mL/g, respectively. At 28 days of aging, mortars S50-10, S30-10, S10-10, and S0-10 had the chemical shrinkage values of 0.13, 0.065, 0.055, and 0.029 mL/g, respectively. Compared to geopolymers S0-10, the chemical shrinkage of the S50-10, S30-10, and S10-10 geopolymers increased by 348.28%, 124.14%, and 89.66%, respectively. Thus, as the slag content increased, the extent of chemical shrinkage increased gradually. These results are consistent with those presented in Section 3.2, that is, the addition of slag promoted shrinkage. As the slag addition promotes the alkali-activated metakaolin reaction and affects the formation of the reaction products, more C-A-S-H gels are formed with increased slag content, thereby increasing chemical shrinkage.

In the first 24 h of curing, the geopolymer paste underwent three stages of volumetric changes, namely, contraction, expansion, and contraction, which are consistent with the results of Li et al. [53]. In the contraction stage, chemical shrinkage occurred due to the dissolution of the metakaolin slag to form monomers or small oligomers. In the subsequent expansion stage, chemical expansion occurred owing to the formation of zeolite-like and aluminum-rich products. In the final contraction stage, the aluminum-rich products recombined with silicate oligomers to form an amorphous silicon-rich gel, resulting in chemical shrinkage. When a sample solidifies under sealed or wet conditions, the metakaolin system expands within a certain curing age [54,55]. Notably, this expansion cannot be explained by the ordinary Portland cement theory, which is usually attributed to ettringite formation or water-absorption-induced expansion.

Figure 6b shows the effect of the Na_2_O content on the chemical shrinkage of the alkali-activated metakaolin-based geopolymer specimens. Increasing the Na_2_O content had a significant impact on the early chemical shrinkage. A large amount of shrinkage was noted in the first 14 days of aging. Thereafter, the Na_2_O content had minimal impact on the chemical shrinkage. At 28 days of aging, the chemical shrinkage values of the S30-8, S30-10, and S30-12 geopolymers were 0.058, 0.065, and 0.068 mL/g, respectively. The chemical shrinkage of the S30-12 geopolymer was 16.55% and 3.63% higher than those of the S30-10 and S30-8 geopolymers, respectively. Thus, chemical shrinkage increased with the increase in Na_2_O content because of the increased dissolution rate of metakaolin slag in a more alkaline environment. These findings correspond well with conclusion drawn in Section 3.2.

The pore structure also affects the geopolymer paste shrinkage [43,56]. Therefore, the drying, autogenous, and chemical shrinkage of the geopolymer pastes were further explored by microscopic analysis, as discussed in the subsequent sections.

### 3.5. FT-IR Analysis

Figure 7a shows the FT-IR spectra of the metakaolin-based geopolymers with different slag contents after 56 days of curing. The spectra contain five clear vibration peaks at 451, 704–718, 865, 1002–1036, and 1645–1650 cm^−1^. The wavenumber of the peaks at 451, 704–718, and 865 cm^−1^, which correspond to the symmetrical vibration of the Si–O–Si functional groups, are inversely proportional to the slag content. This is attributed to the increased consumption of SiO_2_ in the metakaolin-based geopolymers with the increased addition of slag. In addition to generating N-A-S-H gel, the consumed SiO_2_ may also react with the Ca^2+^ ions in the slag to generate calcium-silicate-hydrate (C-S-H) gel.

The peak at 1002–1036 cm^−1^ represents the vibration of the asymmetric T–O–Si (T = Si or Al) functional groups. As the slag content increased from 0 to 50 mass%, the band wavenumber decreased from 1036 to 1002 cm^−1^ because of the promoted alkali-activated metakaolin reaction with the addition of slag. This increased the amount of C-S-H gel in the reaction product and improved the compressive strength of the geopolymers (see Section 3.1). Additionally, the intensities of the bands at 704–718 and 865 cm^−1^, which correspond to the bending vibration of the Al–O–Si bonds, decreased as the slag content increased, which can be related to the slag addition [44,52]. Finally, the wavenumber of the peak at 1645–1650 cm^−1^, which represents the stretching vibration of the H–C–H groups, was inversely proportional to the slag content. This is attributed to the increased water consumption in the paste and promoted the formation of the C-S-H gel as the slag content increased. As the C-S-H gel lost water easily, the vibration peak was weakened. Moreover, the sample shrinkage increased with the slag content, as shown in Section 3.2, Section 3.3 and Section 3.4.

Figure 7b shows the FT-IR spectra of the metakaolin-based geopolymers with different Na_2_O contents after 56 days of curing. The Na_2_O content had a minimal effect on the vibration peaks of the geopolymers, suggesting similar functional groups. When the Na_2_O content increased from 8 to 12 mass%, the T–O–Si vibration peak at 1011–1016 cm^−1^ downshifted, which indicates that the Al substitution rate of Si was higher in a more alkaline environment, that is, the extent of the metakaolin reaction was higher. This confirms that more alkaline environments promote the metakaolin reaction. Therefore, with 8–12 mass% Na_2_O, the compressive strength of the geopolymers increased as the Na_2_O content increased. A sharp peak was noted at approximately 1385 cm^−1^, which is attributed to the stretching vibration of O–C–O, with its intensity increasing as the Na_2_O content increased. This peak is particularly pronounced for the S30-12 mortar because of the reaction of the excess alkali with CO_2_ in air to form carbonates at high Na_2_O contents [57].

### 3.6. SEM Analysis

Figure 8a,b show the morphologies of the metakaolin-based geopolymer specimens with 0 and 50 mass% slag, respectively. The addition of slag resulted in a dense structure after alkali activation. The pure metakaolin-based geopolymers are a flocculated cementitious material with a relatively loose structure, mainly in a block structure (Figure 8a). In contrast, the metakaolin-based geopolymers with 50 mass% slag have a dense structure without layers. This demonstrates that the addition of slag results in tighter packing of the constituent particles. The slag and metakaolin particles have different sizes. After adding slag into the metakaolin-based geopolymers, the slag has a certain filling effect on the metakaolin-based geopolymers. Two minerals (slag and metakaolin) were doped together, and the particles were tightly packed, which reduced the size and number of macropores. A dense and refined effect on the pores of the metakaolin-based geopolymers and an increased density of the hardened structure were noted. With the addition of slag, a large amount of amorphous C-S-H gel was generated after alkali stimulation, which filled the gaps in the formed three-dimensional network structure. This is conducive to the enhancement of the density and compressive strength of the metakaolin-based geopolymers. However, the slag addition also resulted in a large number of microcracks in the metakaolin-based geopolymers, as shown in Figure 8b, which increased the shrinkage degree.

### 3.7. MIP Analysis

To further elaborate the influence of the slag and Na_2_O contents on the pore structure of the metakaolin-based geopolymers, the pore size distribution is shown in Table 3. Based on previous research [58], pores can be classified as: gel pores (<10 nm), transitional pores (10–100 nm), capillary pores (100–1000 nm), and macropores (>1000 nm). Figure 9a shows the effective improvement in the pore size distribution of the metakaolin-based geopolymers with the addition of slag. The pure metakaolin-based geopolymers have large pores (10–100 nm) and a large total pore volume. As the slag content increased, the pore size decreased gradually. With the addition of 50 mass% slag, pores with diameters of less than 10 nm account for the majority of pores (56.60%); however, because of their small size, they only occupy a small volume of the entire geopolymer. Owing to the different particle sizes of slag and metakaolin, combining them effectively reduced the number of pores by improving particle packing. In addition, the hydration products of slag and metakaolin can fuse together to effectively reduce the number of macropores. Consequently, the addition of slag significantly reduced the porosity of the geopolymers. As the slag content increased, the geopolymer porosity, which refers to the percentage of the pore volume in bulk geopolymer materials to the total volume of materials under natural conditions, decreased linearly because of the close packing of the slag and metakaolin particles. The geopolymer without slag has a porosity of 31.45%. When the slag content was 50 mass%, the geopolymer porosity was only 14.22%, which was 54.79% lower than that without slag, and the structural densification degree of the geopolymers significantly improved. These findings are consistent with those reported by Zhu et al. [10]. In addition, this finding can be ascribed to the increased fraction of gel pores with diameters of <10 nm with the slag content, which indicates that more C-A-S-H gels are generated in the system. These C-A-S-H gels undergo pronounced shrinkage, resulting in increased system shrinkage.

Figure 9b and Table 3 show that the addition of alkali can effectively improve the geopolymer pore size distribution, whereby the pore size decreased as the Na_2_O content increases. When the Na_2_O content was 12 mass%, the geopolymer porosity was 24.01% lower than that when the Na_2_O content was 8 mass%. Thus, increasing the Na_2_O content reduced the total geopolymer pore volume owing to the increased reaction degree of the geopolymer under more alkaline conditions. In addition, the highly alkaline environment promoted the dissolution and decomposition of metakaolin and slag particles, resulting in the refinement of the synthetic gel pores, increased fraction of gel pores, increased capillary tension, and increased geopolymer shrinkage.

### 3.8. Relationship between the Drying Shrinkage and Pore Structure Characteristics of the Metakaolin-Based Geopolymers

The shrinkage behavior of the geopolymers is directly related to their pore structure. Figure 10a,b show the effects of different slag and Na_2_O contents, respectively, on the 180-day drying shrinkage and pore structure of the alkali-activated metakaolin-based geopolymer specimens, including the porosity, average pore size, total pore volume, and median pore size. The drying shrinkage and pore structure parameters in Figure 10a,b are defined as the ratio of the corresponding values of the samples to those of S0-10 and S30-8, respectively. As the slag content increased, the drying shrinkage of the metakaolin-based geopolymers increased, whereas the porosity, average pore size, total pore volume, and median pore size decreased. This shows that slag addition reduced the porosity, refined the pore structure, and increased the drying shrinkage. A lower porosity increased the capillary stress in the polymer pore network, thereby increasing the shrinkage strain. As metakaolin has a higher water demand than that of slag, the metakaolin-based geopolymer paste has a higher viscosity, and more pores are introduced during mixing [41]. Therefore, as the slag content increased, the porosity of the paste decreased, and the pore curvature increased, resulting in a decrease in the water loss of the paste. As drying shrinkage is sensitive to water loss, the drying shrinkage increased.

Figure 10b shows that the drying shrinkage of the metakaolin-based geopolymers increased with the Na_2_O content, whereas the average pore size, total pore volume, and median pore size decreased. In addition, increasing the activator concentration had a similar effect to increasing the slag content. Increasing the Na_2_O content increased the drying shrinkage of the metakaolin-based geopolymers and decreased the values of the characteristic pore structure parameters. As increasing the Na_2_O content refined the pore structure of the metakaolin samples containing 30 mass% slag, the pore size refinement [50,56] significantly increased the capillary tension, and consequently, the drying shrinkage. Ma and Ye [59] studied alkali-activated fly ash materials and obtained similar results. The alkali-activated paste with higher sodium silicate and silica contents had a finer pore structure and shrinkage. Ye et al. [14] hypothesized that the activator concentration affects the elastic modulus of alkali-activated slag, thereby affecting its drying behavior. Therefore, in this study, the Na_2_O content had a major impact on the mechanical properties and drying shrinkage of the geopolymers. Combined with previous research, the effects of the Na_2_O content on metakaolin geopolymers is also applicable to other geopolymers [23].

## 4. Conclusions

In this study, the effects of slag and Na_2_O content on the strength and volume stability of alkali-activated metakaolin-based geopolymers were investigated. Based on the results and discussion, the following conclusions can be drawn.

(1)When slag content was 0–50 mass%, the compressive strength of alkali-activated metakaolin-based geopolymers increased with the increases in the slag content. The compressive strength of the metakaolin geopolymer without slag was 70.1 MPa when the curing time was 56 days. When the slag content was 50 mass%, the maximum compressive strength of the metakaolin geopolymer was 86.1 MPa, which was 22.80% higher than that without the slag content. Similarly, the compressive strength increased with the Na_2_O content. The compressive strength of the geopolymers with 12 mass% Na_2_O was 42.36% higher than that of the geopolymers with 8 mass% Na_2_O.(2)When the slag content was 0–50 mass%, as the slag content increased, the drying, autogenous, and chemical shrinkage of the alkali-activated metakaolin-based geopolymers increased. When the Na_2_O content increased from 8 to 12 mass%, the drying, autogenous, and chemical shrinkage of the alkali-activated metakaolin-based geopolymers with 30 mass% slag increased.(3)The slag addition enhanced the formation of the hydration products of the alkali-activated metakaolin and produced a denser gel structure. Thus, a higher Na_2_O content was conducive to the dissolution and decomposition of metakaolin and slag particles, thereby promoting the formation of reaction products.(4)Slag can noticeably improve the pore size distribution of the alkali-activated metakaolin-based geopolymers. As the slag content increased, the porosity of alkali-activated metakaolin-based geopolymers decreased and the gel pores increased, resulting in the increased contraction of the geopolymer. Similarly, the increase in the Na_2_O content reduced the porosity of the alkali-activated metakaolin-based geopolymers, refined the gel pores, and increased the shrinkage of geopolymers.(5)In this paper, the in-depth study on strength and shrinkage of alkali-activated metakaolin-based geopolymers is of great significance for further understanding and utilization of metakaolin-based geopolymers. However, this paper only studied the effects of four levels of slag content (0, 10, 30 and 50%) and three levels of Na_2_O content (8, 10 and 12%) on metakaolin-based geopolymers, which was not enough to draw more conclusions. In the follow-up study, the effects of more slag content and Na_2_O content on the compressive strength and shrinkage of metakaolin-based geopolymers should also be considered.

## Figures and Tables

**Figure 1 materials-15-02944-f001:**
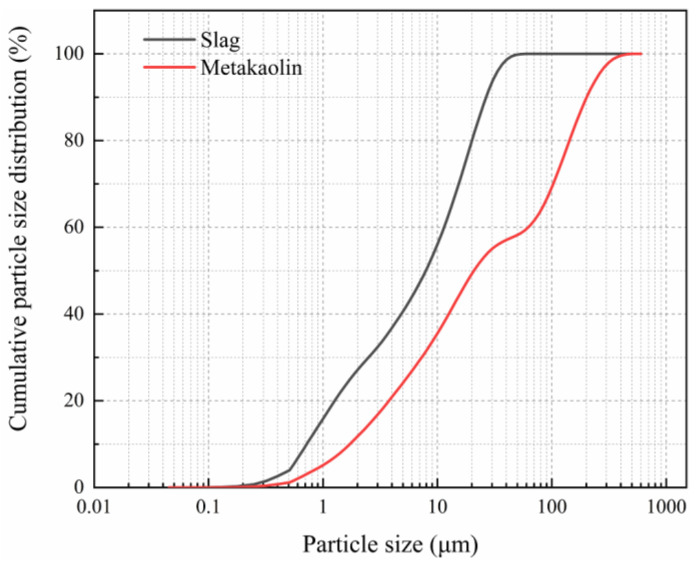
Cumulative particle size distribution of the metakaolin and slag used in this study.

**Figure 2 materials-15-02944-f002:**
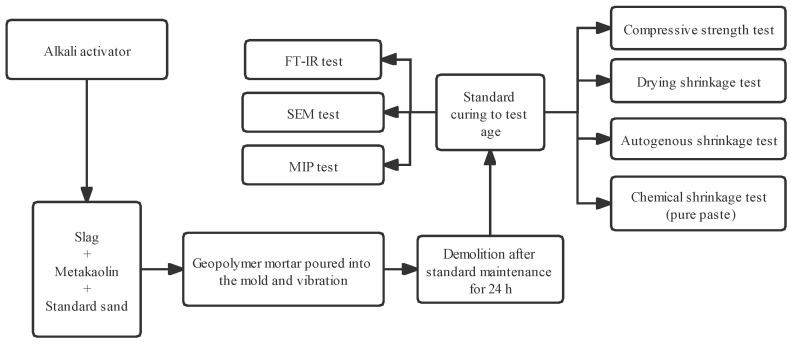
Test process flowchart.

**Figure 3 materials-15-02944-f003:**
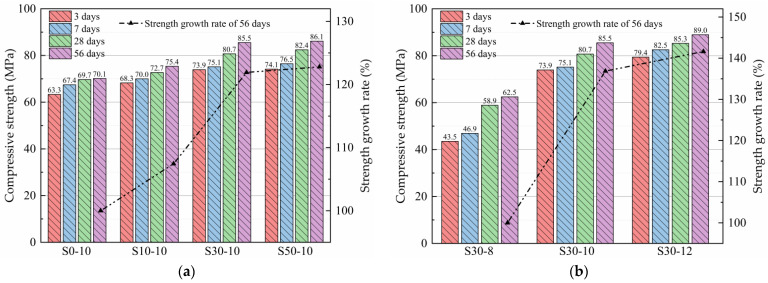
Compressive strength of the alkali-activated metakaolin-based geopolymer specimens according to the (**a**) slag and (**b**) Na_2_O content.

**Figure 4 materials-15-02944-f004:**
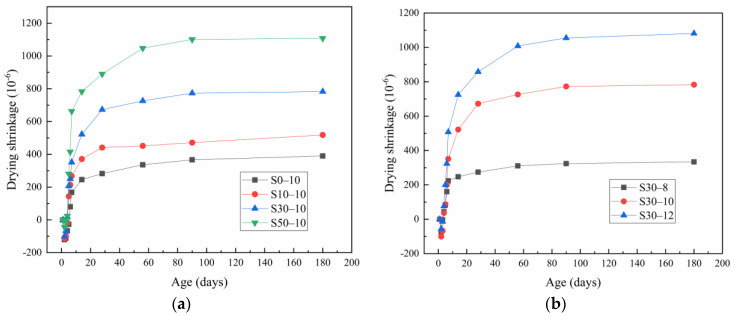
Drying shrinkage of the alkali-activated metakaolin-based geopolymers specimens according to the (**a**) slag and (**b**) Na_2_O content.

**Figure 5 materials-15-02944-f005:**
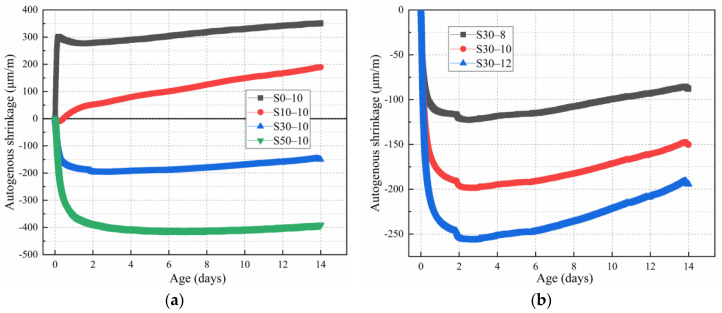
Autogenous shrinkage of the alkali-activated metakaolin-based geopolymer specimens according to the (**a**) slag and (**b**) Na_2_O content.

**Figure 6 materials-15-02944-f006:**
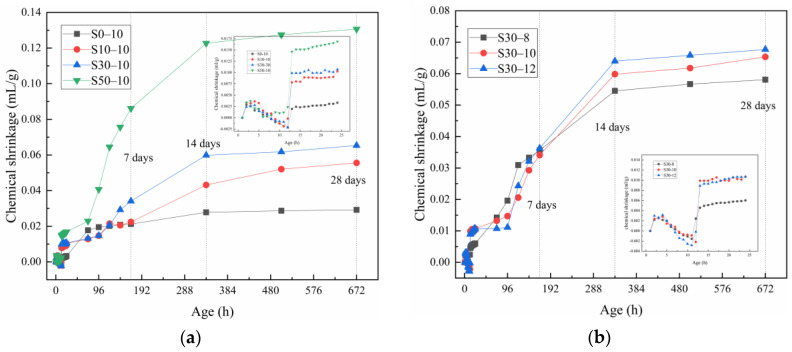
Chemical shrinkage of the alkali-activated metakaolin-based geopolymer specimens according to the (**a**) slag and (**b**) Na_2_O content. The insets show the chemical shrinkage in the first 24 h.

**Figure 7 materials-15-02944-f007:**
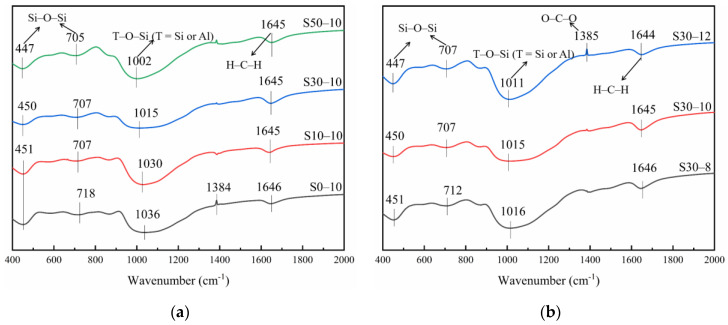
FT-IR spectra of the alkali-activated metakaolin-based geopolymer specimens according to the (**a**) slag and (**b**) Na_2_O content.

**Figure 8 materials-15-02944-f008:**
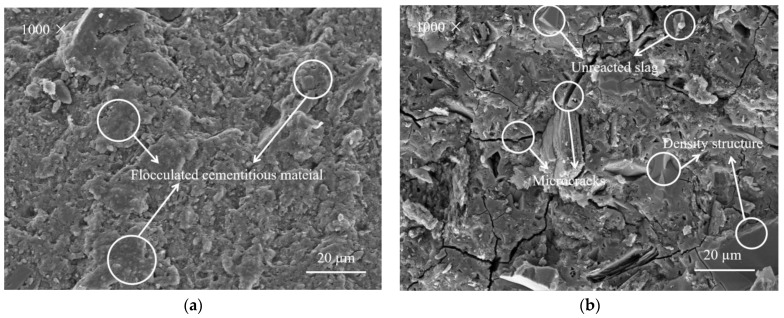
SEM images of the alkali-activated metakaolin-based geopolymer specimens with a slag content of (**a**) 0 mass% and (**b**) 50 mass%.

**Figure 9 materials-15-02944-f009:**
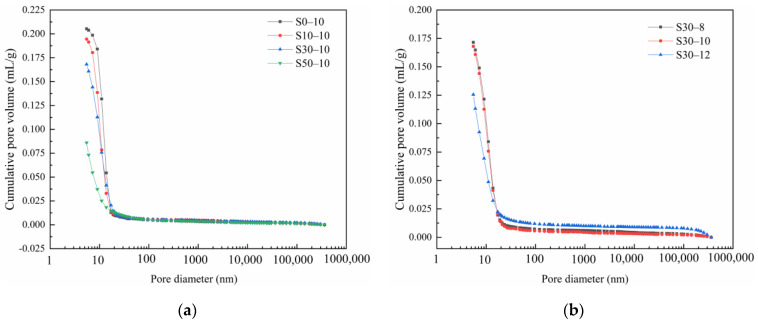
MIP spectra of the alkali-activated metakaolin-based geopolymer specimens according to the (**a**) slag and (**b**) Na_2_O content.

**Figure 10 materials-15-02944-f010:**
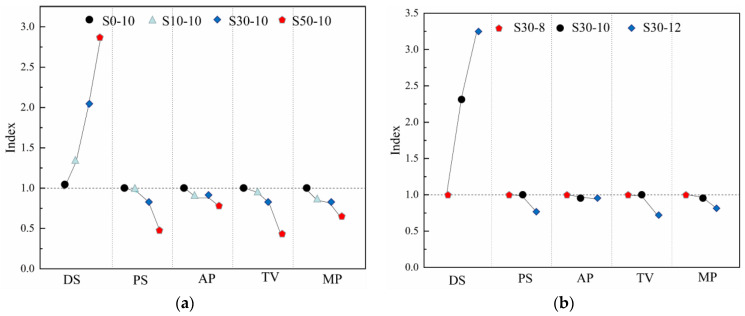
Drying shrinkage and pore structure parameters of the alkali-activated metakaolin-based geopolymers according to the (**a**) slag and (**b**) Na_2_O content.

**Table 1 materials-15-02944-t001:** Chemical composition of the metakaolin and slag used in this study.

Material	Mass Fraction (%)							
	K_2_O	Na_2_O	SO_3_	SiO_2_	Fe_2_O_3_	Al_2_O_3_	MgO	CaO
Metakaolin	0.44	0.41	-	49.78	0.93	34.63	2.58	-
Slag	0.61	0.52	0.24	29.68	3.75	13.46	6.62	36.54

**Table 2 materials-15-02944-t002:** Mixture proportions of the alkali-activated metakaolin-based geopolymers.

**Paste Sample**					
**Mortar Specimen**	**Metakaolin (mass%)**	**Slag (mass%)**	**Modulus**	**Na_2_O (mass%)**	**Water-to Binder Ratio**
S0-10	100	0	1.5	10	0.45
S10-10	90	10		10	
S30-10	70	30		10	
S50-10	50	50		10	
S30-8	70	30		8	
S30-12	70	30		12	
**Mortar Sample**					
**Mortar Specimen**	**Metakaolin (** **g)**	**Slag (** **g)**	**Standard Sand (** **g)**	**Na_2_O (** **g)**	**W** **ater (** **g)**
S0-10	33.3	0	66.7	3.3	15.0
S10-10	29.7	3.3	66.7	3.3	
S30-10	23.3	10.0	66.7	3.3	
S50-10	16.7	16.6	66.7	3.3	
S30-8	23.3	10.0	66.7	2.7	
S30-12	23.3	10.0	66.7	4.0	

Notes: Binder refers to metakaolin + slag, which is equal to 100%. Na_2_O content is the proportion of Na_2_O to the binder. Water refers to deionized water, not the water from the activators. In sample S30-10, S30 refers to the slag content of 30% and 10 refers to the alkali content of 10%.

**Table 3 materials-15-02944-t003:** Pore size distribution of the metakaolin-based geopolymers.

Sample	Porosity (%)	Average Pore Diameter (nm)	Pore Size Distribution (%)			
			<10 nm	10–100 nm	100–1000 nm	>1000 nm
S0-10	31.45	11.87	10.35	87.13	0.86	1.67
S10-10	30.32	10.49	28.75	68.17	0.61	2.47
S30-10	26.06	10.43	32.92	63.69	0.82	2.57
S50-10	14.22	9.08	56.60	37.22	2.00	4.18
S30-8	26.32	10.70	29.11	66.63	0.77	3.49
S30-12	20.00	10.01	44.75	45.71	1.54	8.00

## Data Availability

The data supporting the findings of this study are available from the corresponding author upon reasonable request.

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
