# Peer review of "Effect of Slag on the Strength and Shrinkage Properties of Metakaolin-Based Geopolymers"

_materials, 2022, doi:10.3390/ma15082944_

Round 1
Reviewer 1 Report
Dear Authors, interesting results, but there are a few things that need to be clarified and improved:
Line 34: write please what is the definition of “alkali-activated metakaolin-based geopolymers” - what is the lowest amount of metakaolin in these mixtures or what ranges of composition Si/Al or Na/Si ?
Line 66-67: “…the drying shrinkage de-66 creased by 79.17% to 400 με…” - correct the units
Line 72: “με“ - is the shrinkage unit?
Line 87-91: what kind of slag was used and in what temperature calcined clay was prepared?
Line 97: ”specific gravity and fineness modulus” - some units?
Line 100: modulus: 3.36 – Na/Si? (molar, mass procent?)
Line 105-110: alkali content is not only Na2O content, please accurately describe in the Table 2 what was the composition of the activator - how much water and how much and what reagent was used, then count the Si/Al ratio. There is 100% of metakaolin and 10% of alkali, so we have 110% in S0-10 specimen. Tab 2 must be corrected, without “Alkali content”. “fine aggregate” – a sand? A specimen S50-10 is maybe a slag-based geopolymer? One decimal place is enough (Tab. 2). W/B ratio – water also from activator? Binder is metakaolin + slag? - describe it in detail. How do you know how much active silica is and on what basis did you choose the amount of activator?
Line 116: “Cement mortar strength tests were carried out…” – you tested geopolimer mortars not cement mortars.
Line 117: “The geopolymer paste sample… ” – mortar or paste?
Line 126: “Cement mortar..” – not cement
Check the spelling, for example I noticed an unnecessary dot after "hand" - line 177: “On the other hand. Cui et al. [10,11] came…” or line 489: “with 12 mass% alkali”
Check the editorial requirements, e.g. there should be a greater space between lines 228 and 229
Figure 7 - Assign function groups / bonds to wavenumber values
Figure 8 - The magnification is too high, show images made with 1kx, because there is a slag grain in (b), probably, not a binder
Table 3 – “Porosity” – what kind? Total? And what about Tortuosity? Do you have these values as well?
Line 485: ”… gradually increases with the slag content.” - with decrease or increase of slag content?
Line 492: “The increase in alkali content…” – not increase or decrease, because there was only 1 higher and 1 lower content of alkali amount 8 -10 -12 %
Author Response
Point 1: Line 34: write please what is the definition of “alkali-activated metakaolin-based geopolymers” - what is the lowest amount of metakaolin in these mixtures or what ranges of composition Si/Al or Na/Si ?
Response 1: Line 34 has been resolved, please see lines 37-39.
Point 2: Line 66-67: “…the drying shrinkage de-66 creased by 79.17% to 400 με…” - correct the units.
Response 2: Line 66-67 has been resolved, please see line 86.
Point 3: Line 72: “με” - is the shrinkage unit?
Response 3: Yes, με is the shrinkage unit. please see line 85.
Point 4: Line 87-91: what kind of slag was us ed and in what temperature calcined clay was prepared?
Response 4: Line 87-91 has been resolved, please see line 109.
Point 5: Line 97: “specific gravity and fineness modulus” - some units?
Response 5: Line 97 has been resolved, please see lines 120-122.
Its specific gravity (The specific gravity of the sand is the ratio of the density of the substance (fully dense state) to the density (999.972 kg/m3) at standard atmospheric pressure and pure H2O at 3.98 ℃. It is a ratio, so there is no unit.) and fineness modulus (The fineness modulus is an indicator of the thickness and category of natural sand grain diameter, which has no unit. The greater the fineness modulus, the thicker the sand.)
Point 6: Line 100: modulus: 3.36 – Na/Si? (molar, mass procent?)
Response 6: Line 100 has been resolved, please see line 125.
Point 7: Line 105-110: alkali content is not only Na2O content, please accurately describe in the Table 2 what was the composition of the activator - how much water and how much and what reagent was used, then count the Si/Al ratio. There is 100% of metakaolin and 10% of alkali, so we have 110% in S0-10 specimen. Tab 2 must be corrected, without “Alkali content”. “fine aggregate” – a sand? A specimen S50-10 is maybe a slag-based geopolymer? One decimal place is enough (Tab. 2). W/B ratio – water also from activator? Binder is metakaolin + slag? - describe it in detail. How do you know how much active silica is and on what basis did you choose the amount of activator?
Response 7: Point 7 has been resolved, please see section 2.2.
Thank you for pointing out that alkali equivalent is not just sodium oxide, so I changed it to sodium oxide.
(2) The activators used in this study were prepared by blending sodium hydroxide, waterglass solution, and de-ionized water 24 h before the sample manufacturing.
(3) There is 100% of metakaolin and 10% of alkali, 10% of alkali is proportion of Na2O to binder quality.
(4) Table 2 has been modified, fine aggregate refers to standard sand. A specimen S50-10 can also be called slag-based polymer, but for the sake of full text unification, we call it metakaolin-based geopolymers.
(5) W/B ratio – water is deionized water, not from activators. Binder is metakaolin + slag, notes below table 2.
(6) Silica content in 2.1.3 section manufacturers have given. The amount of silicon oxide is determined according to the modulus of sodium silicate. Selection of activators based on previous studies. For example,
- Yicong Li. Study on the Reaction Level and Influencing Factors of Alkali-Activated GGBFS-Metakaolin Blend. Master thesis, Changsha University of Science and Technology, China, 2020.
- Tao, Yang; H., Zhu; Z, Zhang. Influence of Fly Ash on the Pore Structure and Shrinkage Characteristics of Metakaolin-Based Geopolymer Pastes and Mortars. Constr. Build. Mater. 2017, 153, 284–293. DOI: 10.1016/j.conbuildmat.2017.05.067.
Point 8: Line 116: “Cement mortar strength tests were carried out…” – you tested geopolimer mortars not cement mortars.
Response 8: Point 8 has been resolved, please see lines 147.
Point 9: Line 117: “The geopolymer paste sample… ” – mortar or paste?
Response 9: Point 9 is geopolymers mortar
Point 10: Line 126: “Cement mortar..” – not cement.
Response 10: Point 10 has been resolved. please see section 2.2.
Point 11: Line 126: “Cement mortar..” – not cement.
Check the spelling, for example I noticed an unnecessary dot after "hand" - line 177: “On the other hand. Cui et al. [10,11] came…” or line 489: “with 12 mass% alkali”
Check the editorial requirements, e.g. there should be a greater space between lines 228 and 229.
Response 11: Point 11 has been resolved. please see line 226.
Point 12: Figure 7 - Assign function groups / bonds to wavenumber values.
Response 12: Point 12 has been resolved. please see Figure 7.
Point 13: Figure 8 - The magnification is too high, show images made with 1kx, because there is a slag grain in (b), probably, not a binder.
Response 13: Point 13 has been resolved. please see Figure 8.
Point 14: Table 3 – “Porosity” – what kind? Total? And what about Tortuosity? Do you have these values as well?
Response 14: Porosity refers to the percentage of pore volume in bulk geopolymer materials to the total volume of materials under natural conditions. Tortuosity is derived from a model based on inlet mercury pressure and velocity. I 'm sorry, the mercury intrusion experiment can only get data about pore size distribution, porosity, average pore size, median pore size, total pore volume and so on, without the value of tortuosity.
Point 15: Line 485: “… gradually increases with the slag content.” - with decrease or increase of slag content?
Response 15: Point 15 has been resolved. please see line 550.
Point 16: Line 492: “The increase in alkali content…” – not increase or decrease, because there was only 1 higher and 1 lower content of alkali amount 8 - 10 - 12%.
Response 16: Point 16 has been resolved. please see lines 559-561.

Reviewer 2 Report
The article is about effect of slag on strength and shrinkage properties of metakaolin-based geopolymers. However, some issues must to be addressed:
- Abstract: Please start by expressing the aim of this paper, followed by the rest of the information. Typically, the abstract should provide a broad overview of the entire project, summarize the results, and present the implications of the research or what it adds to its field.
- Please improve the introduction and references section by adding newest articles from 2022 and excluding useless self-citations of the co-authors. The bibliographic foundation is important and well executed, however some new discussions should be inserted, authors should consider some new works in the literature, such as: DOI:10.3390/ma15010202.
- The table 2 raise some doubts: how and why were formulate the percentage choosed for experiments?! Please add scientific justification based on which the authors make this choice presented. Also, explain modulus 1.5 !!!!
- Please enhance the clarity for the sample obtaining procedure: include a new subchapter before 2.3 Test methods.
- The results are merely presented, not properly discussed. Please add explanations for the observed changes. Please give an extended discussion on the obtained results and correlate your findings with previous literature studies and prospective applications.
- More analysis and interpretation of the results should be added for a clearer understanding of observed experimental phenomena.
- The authors must to provide some details about importance of the research and their applicability.
- Please enhance the clarity of the conclusion section in order to highlight the results obtained.
- General check-up and correction of the English language is suggested. There are still some minor typos and grammatical errors.
The author needs to address the abovementioned points for the betterment of the manuscript.
Author Response
Point 1: Abstract: Please start by expressing the aim of this paper, followed by the rest of the information. Typically, the abstract should provide a broad overview of the entire project, summarize the results, and present the implications of the research or what it adds to its field.
Response 1: Point 1 has been resolved, please see abstract.
Point 2: Please improve the introduction and references section by adding newest articles from 2022 and excluding useless self-citations of the co-authors. The bibliographic foundation is important and well executed, however some new discussions should be inserted, authors should consider some new works in the literature, such as: DOI:10.3390/ma15010202.
Response 2: Point 2 has been resolved, please see introduction.
Point 3: The table 2 raise some doubts: how and why were formulate the percentage choosed for experiments?! Please add scientific justification based on which the authors make this choice presented. Also, explain modulus 1.5 !!!!
Response 3: This paper mainly studies the effect of slag content on alkali-activated metakaolin-based geopolymers when slag replaces metakaolin. Binder is metakaolin + slag, and metakaolin content + slag content is 100%. References, e.g.
- Zhenming Li, et al. Mitigating the autogenous shrinkage of alkali-activated slag by metakaolin. DOI: 10.1016/j.cemconres.2019.04.016.2.
- Bo Fu, et al. Understanding the Role of Metakaolin towards Mitigating the Shrinkage Behavior of Alkali-Activated Slag. Materials 2021, 14, 6962.
The modulus is in the water glass, SiO2/Na2O molar ratio. please see section 2.1.3. Modulus is changed by changing the amount of sodium hydroxide.
Point 4: Please enhance the clarity for the sample obtaining procedure: include a new subchapter before 2.3 Test methods.
Response 4: Point 4 has been resolved, please see section 2.2.
Point 5: The results are merely presented, not properly discussed. Please add explanations for the observed changes. Please give an extended discussion on the obtained results and correlate your findings with previous literature studies and prospective applications.
Response 5: Point 5 has been resolved, please see section 2.2. The results of each section later explain the reasons for this phenomenon and have been compared with previous studies.
Point 6: More analysis and interpretation of the results should be added for a clearer understanding of observed experimental phenomena.
Response 6: Point 6 has been resolved, please see section 3.2 and 3.4.
Point 7: The authors must to provide some details about importance of the research and their applicability.
Response 7: Point 7 has been resolved, please see section 3.3 and 3.8.
Point 8: Please enhance the clarity of the conclusion section in order to highlight the results obtained.
Response 8: Point 8 has been resolved, please see section 4.
Point 9: General check-up and correction of the English language is suggested. There are still some minor typos and grammatical errors.
Response 9: Point 9 has been resolved.

Reviewer 3 Report
This aimed to investigate the effect of Slag on the Strength and Shrinkage Properties of Metakaolin-Based Geopolymers. This article gives an insight into the strength and shrinkage properties of Geopolymer composites in the construction sector. However, the article requires revision for further quality improvement. The comments are listed below.
- Abstract: The current problem and the solution for the problem should be highlighted in the manuscript.
- Abstract: The work description is unclear and should be discussed more in detail.
- Why has the slag content chosen (0, 10, 30, and 50 mass%) and 40% not been considered?
- Include more literature on the strength and shrinkage properties of geopolymer. The literature section is very weak and it should be improved by adding more literature.
- Mention the size of sand used in this investigation and include the particle size distribution graph.
- The material and method section is shallow and it should be improved. Also, section 2.2 should be discussed in more detail.
- Table 2, on what basis the metakaolin and alkali content was decided? Explain this in the manuscript.
- Section 2.3.2 provides a citation for this standard “ISO GB/T 17671-1999” and other similar standards mentioned in other sections of the manuscript.
- Section 2.3.2, is the sample is geopolymer paste or geopolymer mortar?
- Figure 3 (b), mixture id “S30-10” not defined in Table 2. Please check.
- Figure 10, how does the index is calculated?
- The conclusions are presented with key findings.
- English language should be checked for the abstract and introduction sections.
Author Response
Point 1: Abstract: The current problem and the solution for the problem should be highlighted in the manuscript.
Response 1: Point 1 has been resolved, please see abstract.
Point 2: Abstract: The work description is unclear and should be discussed more in detail.
Response 2: Point 2 has been resolved, please see abstract.
Point 3: Why has the slag content chosen (0, 10, 30, and 50 mass%) and 40% not been considered?
Response 3: Because when the content of slag is similar, the effect of slag on alkali-activated metakaolin-based geopolymers is small. The content of 0, 10, 30 and 50% slag is selected to make the test results more obvious under different slag content.
Point 4: Include more literature on the strength and shrinkage properties of geopolymer. The literature section is very weak and it should be improved by adding more literature.
Response 4: Point 4 has been resolved, please see introduction.
Point 5: Mention the size of sand used in this investigation and include the particle size distribution graph.
Response 5: Sand production standard conforms to the GB / T 17671-1999 ' cement mortar strength test method ' specification. Standard sand is quartz sand that meets the standard after processing. The size range of standard sand is 0.25-0.5 mm. The picture is as follows :
Point 6: The material and method section is shallow and it should be improved. Also, section 2.2 should be discussed in more detail.
Response 6: Point 6 has been resolved, please see section 2.2.
Point 7: Table 2, on what basis the metakaolin and alkali content was decided? Explain this in the manuscript.
Response 7: This paper mainly studies the effect of slag content on alkali-activated metakaolin-based geopolymers when slag replaces metakaolin. Binder is metakaolin + slag, and metakaolin content + slag content is 100%. References, e.g.
- Zhenming Li, et al. Mitigating the autogenous shrinkage of alkali-activated slag by metakaolin. DOI: 10.1016/j.cemconres.2019.04.016.
- Bo Fu, et al. Understanding the Role of Metakaolin towards Mitigating the Shrinkage Behavior of Alkali-Activated Slag. Materials 2021, 14, 6962.
Percentage of alkali content refers to the ratio of sodium oxide content to binder content (alkali content is Na2O/binder). Binder is metakaolin + slag, and metakaolin content + slag content is 100%. please see section 2.2 and notes in table 2.
Point 8: Section 2.3.2 provides a citation for this standard “ISO GB/T 17671-1999” and other similar standards mentioned in other sections of the manuscript.
Response 8: Point 8 has been resolved, please see references.
Point 9: Section 2.3.2, is the sample is geopolymer paste or geopolymer mortar?
Response 9: Point 9 has been resolved, the sample is geopolymers mortar.
Point 10: Figure 3 (b), mixture id “S30-10” not defined in Table 2. Please check.
Response 10: Point 10 has been resolved. please see notes in table 2.
Point 11: Figure 10, how does the index is calculated?
Response 11: Point 11 has been resolved. please see section 3.8.
Point 12: The conclusions are presented with key findings.
Response 12: Point 12 has been resolved. please see the conclusions.
Point 13: English language should be checked for the abstract and introduction sections.
Response 13: Point 13 has been resolved.

Round 2
Reviewer 1 Report
Thanks for taking into account the propositions.
Good luck!
Reviewer 2 Report
The article is suitable for publication, after all improvements.
Reviewer 3 Report
Nil